# Understanding the Online Environment for the Delivery of Food, Alcohol and Tobacco: An Exploratory Analysis of ‘Dark Kitchens’ and Rapid Grocery Delivery Services

**DOI:** 10.3390/ijerph19095523

**Published:** 2022-05-02

**Authors:** Chiara Rinaldi, Marlene D’Aguilar, Matt Egan

**Affiliations:** 1Department of Public Health, Environments and Society, London School of Hygiene and Tropical Medicine, London WC1H 9SH, UK; matt.egan@lshtm.ac.uk; 2Public Health, Haringey London Borough Council, London N22 8HQ, UK; marlene.daguilar@haringey.gov.uk

**Keywords:** online food environment, dark kitchens, online food delivery, grocery delivery, online tobacco sale, online alcohol sale

## Abstract

Online spaces are increasingly important in the sale of food, alcohol and tobacco. This analysis focuses on two developments in online food delivery: delivery-only ‘dark kitchens’ and rapid grocery delivery services (RGDS), with the aim to understand and assess the availability of health harming and health promoting products through these services. Data was collected for one metropolitan local authority in London, UK, using publicly available online sources. Being explorative in nature, the analysis includes descriptive statistics and qualitative assessment. Three dark kitchens (renting kitchens to 116 food businesses), three grocery delivery apps, and 76 grocery businesses available through online delivery platforms were identified. Most businesses renting dark kitchen space were ‘virtual restaurants’ (52%) selling fast food (47%) or dessert (21%) through online delivery platforms. RGDS sold a variety of items, with a focus on pre-packaged foods high in fat, salt and sugar, alcoholic beverages and tobacco. These items were also most likely to be promoted through offers and promotional language. Fruits and vegetables were less commonly available and mainly on grocery delivery apps. Online delivery services increase the temporal and geographic availability and promotion of many unhealthy products. Research expanding on the geographic area of interest is needed.

## 1. Introduction

Non-communicable diseases (NCDs) such as cardiovascular disease, cancers and diabetes are the leading cause of premature mortality in England [1]. Diets high in salt, trans fats and refined sugar, and low in fruits and vegetables, are major risk factors for many NCDs. Diets are known to be determined by environmental factors, such as the availability, accessibility and affordability of harmful commodities [2,3]. The wide availability of ‘fast food’ at a low price contributes to a so-called ‘obesogenic’ environment conducive to unhealthy diets. In England, fast food outlets tend to be concentrated in the most socio-economically deprived areas [4,5,6]. Furthermore, harmful commodities such as alcohol and tobacco are often sold alongside food associated with poor diets. There has been some interest in how these different ‘product environments’ co-exist and interact in ways that influence consumption and (adverse) health outcomes [5,7].

While physical environments are relevant for people’s health behaviors, it is becoming increasingly important to consider how online spaces support or hinder healthy diets and other forms of consumption [8,9,10]. Online (or digital) food environments are defined as “the online settings through which flows of services and information that influence people’s food and nutrition choices and behavior are directed” [10] (p. 1). This includes food retail, digital marketing of food and the influence of social media on eating behavior. In recent years, food purchasing has increasingly shifted to online spaces. In 2019 alone an estimated 900,000 additional meals were sold online each week by restaurants in London, UK [11]. This number has likely increased since due to impacts of the COVID-19 pandemic and the associated temporary closure of hospitality venues [9,12,13,14,15]. Many online food delivery platforms also sell alcoholic beverages and tobacco products [10].

Online food ordering and delivery predominantly takes place through third-party food delivery platforms (e.g., Deliveroo, Just Eats, and Uber Eats), which are exclusively accessible online through their websites or mobile applications (also known as ‘apps’). Online food delivery platforms partner with local food businesses to collect and deliver meals to the customer [16]. Orders are delivered by the platforms’ delivery ‘riders’. Customers choose, order and pay for their meals online through the platform. In this way, online delivery platforms make consuming meals away from home more convenient while also increasing access to a wider variety of food outlets outside of the local area [15]. Both self-reported convenience and greater access to food outlets are associated with increased use of online food delivery [17,18,19,20]. Similar to ‘fast food’ take-away meals, meals ordered online tend to be high in fat, salt and sugar and low in whole grains, fruits and vegetables [15,21,22,23,24,25]. Frequent consumption of these types of meals has been associated with poorer diets [25,26,27], greater odds of overweight and obesity [20,25] and higher cardio-metabolic [28] and cardiovascular risk factors [27,29]. Concerns have also been raised about the impacts of food delivery on the environment (i.e., high carbon footprint, waste generation) and traffic systems (i.e., traffic congestion, road accidents), which could affect public health indirectly [30,31]. In addition to meal delivery, rapid online grocery delivery services have also gained popularity [32]. These services aim to deliver selected food items within minutes from ordering and thereby increase and facilitate access to a variety of healthy and unhealthy products.

The digital sphere brings both challenges and opportunities for public health globally, as has recently been acknowledged by the World Health Organization (WHO office in Copenhagen, Denmark), many of which remain under researched [9,10,15,31]. A deeper understanding of how different online spaces influence eating behavior, especially among the younger generations, is necessary for the development of appropriate policy responses. As the digital food environment is continuing to change and expand with new platforms and services, the boundary between traditional and ‘online’ food businesses is at times blurred. This analysis will specifically focus on two recent developments in online food delivery that are likely to increase access to products that impact on health: ‘dark kitchens’ and rapid grocery delivery services (RGDS) [32,33]. Academic literature specifically focusing on these services tends to be in the fields of business administration [34,35,36] and consumer behavior [37,38,39]. Dark kitchens and RGDS have not yet been studied from a public health perspective, despite the concerns raised in the public health literature [12,18,40] and by public health practitioners who were involved in the conceptualization of this analysis. 

### 1.1. Dark kitchens

Dark kitchens (also known as ‘cloud kitchens’ or ‘ghost kitchens’) are delivery-only commercial kitchens that rent out shared or private kitchen spaces to food businesses [34]. Dark kitchen tenants include fast food restaurants and takeaways wishing to widen their delivery span without the costs of opening an additional high street restaurant [30]. Other tenants are food entrepreneurs responding to the increased demand for food delivery by opening ‘virtual restaurants’ that can only be accessed through online delivery platforms. Dark kitchens have been particularly successful in the United Kingdom during the COVID-19 pandemic. According to Deliveroo, one of the large dark kitchen operators in the UK, there has been a 70% increase in the average order volume per dark kitchen since March 2020 [33]. The rise of dark kitchens is an international phenomenon, but dark kitchens are particularly common in urban areas with high population density [34,37]. It is believed that dark kitchens could undermine efforts by local governments to curb the growth of fast food outlets through policy levers such as urban planning restrictions [18,40].

### 1.2. Rapid Grocery Delivery Services

Various types of online RGDS exist. Unlike traditional supermarkets with a high street presence, new grocery delivery businesses operate from centrally located warehouse spaces that are not open to the public. Customers use a mobile app to browse and order grocery items online and track their delivery [35]. These app-based businesses will also be referred to as ‘dark grocers’. Examples are Gorillas, Zapp and Getir. Since 2020, dark grocers are available in most areas of London and several other UK cities [32,41].

Rapid grocery delivery is also offered by traditional grocers in partnership with external food delivery platforms such as Deliveroo and Uber Eats [42]. Despite operating physical shops, grocers that make use of delivery platforms share similarities with app-based dark grocers—for example in the way that food is ordered and the rapid delivery.

### 1.3. Scope of This Paper

This analysis explores the presence and possible public health impacts of dark kitchens and online RGDS in one metropolitan local authority in London, United Kingdom. The aim is to understand and assess the availability of health harming and health promoting products through dark kitchens and RGDS. We assessed where these services are located, how these services work, and how they impact on the local availability of products known to affect health, focusing primarily on food, alcoholic and non-alcoholic beverages and tobacco. Understanding how dark kitchens and RGDS add to the local food environment is relevant for policy development in public health and related areas. The research was guided by the following questions: (1) What dark kitchens and RGDS are available? (2) How do dark kitchens and RGDS affect availability of harmful commodities, including food, alcohol and tobacco? (3) How are products and services of dark kitchens and RGDS services advertised?

## 2. Materials and Methods

### 2.1. Setting

The setting for this research is the London Borough of Haringey, a local authority in North London. The research has been conceptualized in collaboration with the Haringey Council Public Health team. According to the latest census, Haringey has over 254,900 inhabitants from a variety of ethnic and socio-economic backgrounds [43]. While Haringey is the sixth most deprived borough of London, there is a stark disparity between wards. Some areas in the east of Haringey are classified as among the most deprived of England, while the areas in the west are among the least deprived [44]. This is also reflected in the food environment. According to a 2018 report by Public Health England, Haringey has 77.2 high street fast food outlets per 100,000 people, which is below the England average of 96.1 [45]. However, fast food outlets (including burger bars, kebab, chicken and chip shops and pizza outlets) tend to be more prominent in more deprived wards. Research on Haringey’s online food environment does not yet exist.

### 2.2. Data Collection

All information was identified online in September–November 2021 and is available in the public domain. Other studies have analyzed various aspects of online food delivery apps using similar approaches [13,21,24].

#### 2.2.1. Dark Kitchens

An online search using Google search engine was conducted to identify dark kitchens with physical premises in the London Borough of Haringey. Search terms used include ‘dark kitchens’, ‘cloud kitchens’, ‘ghost kitchens’ in London or Haringey. Dark kitchens in the surroundings of Haringey were not included in this analysis, as it is not always clear from the data sources whether these businesses deliver to (parts of) Haringey. Information about the kitchen spaces was collected using company websites. Data on the location of the dark kitchens; the equipment and services available; and the types of businesses renting dark kitchen space were collected. Dark kitchen tenants were identified through the Food Standards Agency website [46] and online delivery platforms Deliveroo, Uber Eats and Just Eats. Searches on online delivery platforms were limited to the London Borough of Haringey. A filter for food businesses that deliver from Deliveroo dark kitchens was used on the Deliveroo platform. This filter was not available on other platforms.

The information collected about dark kitchen tenants includes the type of food business; food and alcoholic beverage offer; the service used for delivery; food hygiene ratings and hours of service. Food businesses were coded with one primary food type or cuisine based on their online food menu to facilitate data analysis. ‘Fast food’ was defined using Public Health England’s definition of “food that is energy dense and available quickly, therefore it covers a range of outlets that include, but are not limited to, burger bars, kebab and chicken shops, chip shops and pizza outlets” [45].

#### 2.2.2. Rapid Grocery Delivery Services

Two types of RGDS were analyzed: (1) app-based ‘dark grocers’ and (2) grocery businesses available via online delivery platforms Deliveroo and Uber Eats. These services were identified through Google search engine and the Apple App Store. Apps of national supermarket chains or meal kit delivery services were excluded from this analysis. Information about the grocery delivery services was collected using the service’s website and/or app. This includes information on the location of the business; food hygiene ratings; hours of service; food options; alcohol and tobacco sales; and advertisement and promotion strategies. Food prices of online food delivery platforms were compared to those of three UK supermarket chains. We selected 33 items for the price comparison, which were taken from the Grocer 33 list [47]. The list, produced by The Grocer, contains 33 products that are part of the ‘standard’ shopping basket in the UK and has previously been used to analyze the online food environment [47]. Specific products brands and package sizes were selected where they were not specified on the list to allow for a more accurate price comparison (see Appendix A).

The delivery location for grocery delivery services was set for the London Borough of Haringey or a centrally located address in the Borough. Grocery apps were accessed on an Apple iPhone. Other RGDS were accessed via the Deliveroo and Uber Eats websites. The searches were carried out without an account log in to avoid bias through cookies or saved shopping history.

### 2.3. Data Analysis

All extracted data was compiled into an Excel spreadsheet. As this work is explorative, the analysis was descriptive in nature. We iterated five broad themes to aid analysis and presentation of data: businesses and platforms; geographic and temporal availability; food offer; alcohol and tobacco offer; and promotions and advertising. The analysis included descriptive statistics intended to map the online environment under investigation, and (particularly with promotions and advertising) more qualitative assessments. Dark kitchens and RGDS were analyzed separately within each theme. Dark grocer apps and grocery businesses on delivery platforms were separated at the data analysis stage due to differences in the data (e.g., data on the contents of an app as a whole versus data on multiple individual grocery businesses). Percentages were calculated for variables including the types of food businesses (e.g., virtual restaurants, restaurant chains) and food and alcohol offer. For dark grocer apps, the total amount of alcoholic beverages and tobacco products were calculated. Other information about the identified businesses, including hours of service and delivery times were described qualitatively. For RGDS, price comparisons were made calculating the price of the total shopping basket (based on the Grocer 33 list).

### 2.4. Practitioner and Researcher Informants

Informal, unstructured consultations were held with practitioners and researchers to gather information about the online food environment, and the challenges and opportunities it brings from a variety of perspectives. These were used to inform the data collection. Practitioners from various fields were consulted, including public health, economic regeneration, employment, urban planning, tobacco control, trading standards and environmental health. To comply with confidentiality and data sharing requirements, practitioners were asked to only share information that is in the public domain.

## 3. Results

### 3.1. Businesses and Platforms

#### 3.1.1. Dark Kitchens

Three dark kitchens were identified within the London Borough of Haringey: Karma Kitchen, Deliveroo Editions, and Foodstars. Besides fully equipped private or shared kitchen spaces, dark kitchens also offer services including waste management, cleaning, pest control and business support programs. Summary information about the dark kitchens and the equipment and services they make available to tenants is summarized in Table 1.

In total, 124 dark kitchen tenants were identified at the time of data collection. Eight food businesses could not be further identified online and were therefore excluded from the analysis, leaving 116 businesses. This number could be an underrepresentation as dark kitchens are also available to food brands that may be harder to identify online. The majority of businesses using dark kitchens were virtual restaurant brands (52.6%), and another 7.8% were virtual bakeries. Eight of the 14 virtual restaurants in Deliveroo Editions were operated by Deliveroo itself. Other food businesses included national restaurant/takeaway chains (19%), independent restaurants/takeaways (5.2%), catering businesses (4.3%) and food stalls (1.7%). ‘Other’ food businesses—including (frozen) meal subscription services and food product brands—made up 9.5% of all food businesses.

#### 3.1.2. Rapid Grocery Delivery Services

Three dark grocers were found to be active in the London Borough of Haringey at the time of data collection: Getir, Gorillas and Zapp. At the time of data collection, Getir had four warehouses in Haringey, while the other businesses each had one. More information about the identified dark grocer apps and the products they offer can be found in Table 2.

The other RGDS that were analyzed are those through online delivery platforms Deliveroo and Uber Eats. At the time of data collection, 81 businesses that deliver groceries to Haringey were identified (54 on Deliveroo, 27 on Uber Eats). Two businesses were available on both platforms and were only included once in this analysis. Another three businesses were excluded due to selling meals instead of grocery items, leaving 76 unique businesses.

About a third (35.5%) of the identified grocery businesses were virtual alcohol shops, businesses specializing in the sale of alcoholic beverages that are only available online through delivery platforms. Another 14.5% of the businesses were off-license shops predominantly selling alcoholic beverages and a limited selection of food items (e.g., confectionary items and savory snacks). Convenience stores made up 19.7% of the food businesses, followed by supermarket chains (10.5%) and independent supermarkets (9.2%). The rest of the businesses (10.6%) were butchers, delicatessen shops, a bakery, a petrol station and a cheese shop.

### 3.2. Geographic and Temporal Availability

#### 3.2.1. Dark Kitchens

The majority of businesses using dark kitchens made their food available through online food delivery platforms. With the exception of the food businesses in Deliveroo Editions (which are only available on Deliveroo), most food businesses used multiple platforms. 79.3% of the identified food businesses were available through Deliveroo, 37.9% through Uber Eats and 16.4% through Just Eat. Delivery platforms do not set a maximum distance or minimum delivery time from food outlet to the delivery address (this depends on factors such as the food type, delivery area and the business at the time of delivery [51]). About a quarter (26.7%) of food businesses (also) accepted orders through their own website. A small number of businesses used nationwide postal services for a less rapid delivery (e.g., some virtual bakeries).

Hours of service were only identified for businesses available on online delivery platforms, but appeared to vary slightly across different platforms. Most food businesses were only available during the evening (±17:00–23:00). Fourteen businesses in Karma Kitchen were available for orders past 23:00, of which four until at least 04:00. A minority of businesses was also available in the afternoon (from 12:00) or morning (from 10:00). Opening days varied and it was common for food businesses to be closed at least one day a week.

#### 3.2.2. Rapid Grocery Delivery Services

Getir currently has four warehouses in Haringey, Gorillas and Zapp only have one and are therefore less widely available within the Borough. Orders from dark grocers are exclusively made through their apps, after which the order is prepared and delivered from the local warehouse. According to the apps, products are delivered within ten (Gorillas and Getir) or 20 min (Zapp) from ordering. App-based services operate seven days a week with extended opening hours, ranging from 8:00 to mid-night (Gorillas and Getir) to 24 h a day (Zapp). The price for delivery ranges from £1.80 (Gorillas) to £1.99 (Zapp and Getir).

Orders from grocery businesses on delivery platforms are prepared by the grocery business but delivered by the platform through which they are ordered. Stated delivery times for grocery businesses using delivery platforms such as Deliveroo and Uber Eats range between 20 to 60 min. Most of the grocery businesses identified as providing rapid deliveries for Haringey were located within the Borough, although premises situated outside the Haringey boundaries also offered deliveries to (parts of) Haringey. For example, some virtual alcohol shops are located in self-storage facilities across London, which they use as warehouse spaces for their products.

Orders on delivery platforms can usually only be made during the opening hours of the grocery business (although some businesses allow for pre-orders). Since retailers on the online delivery platforms have physical shops, the hours of service correspond to those of ‘traditional’ shops. Convenience stores and off-licenses typically had longer hours of service online than supermarkets, and were found to deliver until 23:00 (and up to 03:00). Virtual alcohol delivery shops were only available at night time, typically from 21:00–5:00. One business was open 24-h a day.

Delivery fees varied from £0.99–£3.49 on Deliveroo and £0.79–£3.29 on Uber Eats. Alcohol delivery shops had higher delivery fees, ranging from £4.99 to £9.99 on Deliveroo. Grocery businesses on Deliveroo had a minimum order value of £7–£15 or £25 for alcohol delivery shops.

### 3.3. Food Offer

#### 3.3.1. Dark Kitchens

The vast majority of food businesses using dark kitchens sold hot meals for online delivery. A variety of different food types and cuisines were identified. As can be seen in Table 3, the most popular food options were burgers (20.7% of businesses), Italian (12.1%) and (fried) chicken (12.1%).

Almost half (46%) of the identified food businesses were classified as selling ‘fast food’ (including burger bars, fried chicken outlets and some pizza outlets). Two thirds (64.2%) of businesses selling fast food were virtual restaurants. However, most food businesses not classified as primarily selling ‘fast food’ were still found to serve food that is high in fat, salt and/or sugar. The most popular non-fast food options in Haringey’s dark kitchens were dessert foods (11.2%), Indian cuisine (6%) and Japanese cuisine (5.2%) (Table 3).

#### 3.3.2. Rapid Grocery Delivery Services

RGDS sell a variety of food and non-food items, including fruits and vegetables, cupboard items, confectionery, alcoholic beverages, tobacco and e-cigarette products, household cleaning items, personal care, and over-the-counter medication. On the three grocery apps, different categories were displayed on the front page, which appears as soon as the app is opened. The order and way in which the grocery categories are presented could influence how users navigate the app and consume, similar to the lay-out of physical supermarkets [52]. Whilst the product types offered across the three apps were broadly similar, Gorillas had the greatest variety of fresh fruit, vegetables and herbs (*n* = 85), compared to Zapp and Getir (*n* = 44 and *n* = 38, respectively). Fresh fruit and vegetables were displayed among the first food categories on Zapp and Gorillas, making them more visible to customers. On Getir, fruits and vegetables were shown after confectionery and alcohol (i.e., requiring more scrolling down the screen).

When it comes to grocery businesses on delivery platforms, some businesses (namely virtual alcohol shops and some off-licenses) did not sell food at all. Only a quarter of grocery businesses (25%) sold fruit and vegetables online. This included all supermarket chains, four independent supermarkets and seven convenience stores. Supermarket chains had the greatest fruit and vegetable offer. However, the offer was limited compared to the availability in-store or on supermarkets’ own websites.

Food prices were relatively high on all analyzed RGDS. On average, a sample shopping basket was £86.13 when bought from dark grocers compared to £75.23 from supermarket chains (14.5% price difference). The prices of supermarkets order made through Deliveroo and Uber Eats were also higher than those in-store. The difference was £13.96 (20%), £7.72 (11.4%) or £4.33 (7.7%) depending on the supermarket.

### 3.4. Alcohol and Tobacco Offer

#### 3.4.1. Dark Kitchens

Nineteen (16.4%) of the identified food business sold alcoholic beverages through their dark kitchen location, although this differed per dark kitchen. Thirteen food businesses using Deliveroo Editions and six using Karma Kitchen sold alcohol, while no alcohol was sold by businesses using Foodstars’ dark kitchen. Beer (*n* = 17) and wine (*n* = 6) were most frequently available. Businesses using dark kitchens were not found to sell tobacco and e-cigarette products.

#### 3.4.2. Rapid Grocery Delivery Services

Dark grocer apps had a large assortment of alcoholic beverages (Zapp, *n* = 279; Gorillas, *n* = 274; Getir *n* = 196). Most of the products for sale were either ‘spirits’ (28.4% of alcoholic beverages), ‘wine and champagne’ (28.4%), and ‘beer and cider’ (27.4%) (Figure 1). Ready to drink cocktails and mixed drinks accounted for 12% of the total alcohol offer. Only 4% of the alcohol offer were non-alcoholic alternatives (0% ABV). Dark grocers also had a relatively large tobacco selection. Zapp sold the largest variety of tobacco and e-cigarette products (*n* = 50), compared to Gorillas (*n* = 36) and Getir (*n* = 33) (Figure 2). Cigarettes and rolling tobacco, and e-cigarettes were equally represented in the tobacco category (38.7%). Non-combustible tobacco products (10.1%) and tobacco-free nicotine pouches (9.2%) were less commonly available.

Of the 76 identified grocery businesses on delivery platforms, the vast majority (90.8%) sold alcoholic beverages online, of which about half exclusively or primarily sold alcohol. The majority of businesses with alcohol on their online menu sold spirits (97.1%), wine and champagne (95.7%) and beer and ciders (94.2%). Only 20.3% of the businesses also sold non-alcoholic alternatives. In addition, over half (61.8%) of the grocery businesses on delivery platforms sold tobacco and/or e-cigarettes. Cigarettes and rolling tobacco were more likely to be sold by supermarket chains, convenience stores and off-licenses. Virtual alcohol shops were in turn more likely to sell e-cigarettes.

When it comes to product visibility, alcohol was prominently featured on the front page of all apps and on the online menus of virtual alcohol shops and off-licenses. Tobacco products were less visible on dark grocer apps. Supermarket chains and independent supermarkets available on delivery platforms made alcoholic beverages and tobacco less visible by placing them at the bottom of their online menu, requiring scrolling down.

Something to note is that many businesses on delivery platforms (particularly virtual alcohol shops) did not display images for their tobacco and e-cigarettes. While this could make these products less visible on an online menu, it also removes the health warnings that are present on these products by law, including text and images. Businesses did not otherwise include a health warning on tobacco products (e.g., in the product description or at point of order). How alcohol and tobacco is displayed and marked also depends on the online food delivery platform. Alcoholic beverages were only marked on Uber Eats (with ‘Alcohol’ or ‘Contains alcohol’). Both Deliveroo and Uber Eats specify at point of order that alcohol and tobacco can only be sold to customers aged 18+ years but ID checks are performed at point of delivery. Dark grocer apps clearly showed the health warnings present on tobacco packaging. On Getir, alcoholic beverages and tobacco products contained an age restriction and health warning in the product description. Zapp and Gorillas had a simple age verification for accessing and purchasing alcohol and tobacco. Identification for age verification purposes is only required at point of delivery.

### 3.5. Promotions and Advertisement

#### 3.5.1. Dark Kitchens

Dark kitchens tended to target their promotional activities at food businesses (i.e., potential tenants) rather than consumers. However, food businesses renting space in dark kitchens promoted food and drink offers directly to consumers, for example on their websites, social media or through online delivery platforms. Deliveroo’s platform promoted ‘partnerships’ that allowed consumers to purchase products from more than one business at the same time. For example, certain fast food businesses using Deliveroo Editions partnered with a brewery or bakery to promote orders of alcoholic beverages or desserts alongside a meal order.

#### 3.5.2. Rapid Grocery Delivery Services

RGDS are directly advertised to the end consumer. Non-monetary promotions were the most common type of promotion on dark grocer apps. These include the products appearing on the first page of a category, ‘recommended products’ categories, and products shown on banners at the top of the app. Price promotions were less frequent.

Promotions tended to focus on foods high in fat, salt and sugar, and alcohol. On Getir, banners featured price discounts on ice-cream, seasonal sweets (Halloween) and alcoholic beverages, but also dairy-free milk alternatives and fruit smoothies. The Zapp app had banners with seasonal treats (‘Zappy Halloween’), ‘Afternoon snacks’ (mostly consisting of sweet snacks and sugar-sweetened beverages), and over-the-counter medication and supplements (‘Flu season savers’). Gorillas was the only app to not promote specific food items or brands on their front page banners. The app was also more likely to show images of fresh food and vegetables. All apps included free delivery offers in their banners.

The Gorillas app had a special section dedicated to recommended products (‘Gorillas recommends’), including seasonal sweets and confectionary items, alcoholic beverages, ready meals, and new items on the app. The only app offering special price promotions on products was Getir. Their ‘Special offers’ category featured discounted items in all grocery categories except for fresh fruits and vegetables. About half of the discounts were on non-food items, and 12 were on alcohol (including combination deals on alcoholic beverages and snacks).

Grocery businesses on Deliveroo and Uber Eats similarly used non-monetary promotions, including the products shown on their ‘cover image’ and at the top of their online food menu. Again, the items that were more likely to be promoted tended to be high in fat, salt and sugar, or alcoholic. As shown in Table 4 over two thirds (69.7%) of the cover images of grocery businesses included (or exclusively contained) alcoholic beverages. Images also featured confectionery items (31.6%); salty snacks (19.7%) and sugar-sweetened beverages (19.7%). Other common products were cupboard items (19.7%), including cereals, canned legumes and flours; meat and alternatives (14.5%); and dairy and eggs (11.8%). A small proportion of businesses, typically supermarkets, promoted fruit and vegetables (7.9%) in their cover image.

Given that a large amount of the identified businesses were (virtual) alcohol shops, alcoholic beverages were most likely to be at the top of the online food menu. This was followed by categories with special promotions and ‘bundles’ (e.g., ‘PARTY PACK OFFER’, ‘Pandemic specials’, ‘Flash deal’). Cupboard items and seasonal categories (e.g., BBQ and ‘Organic September’) were commonly featured by supermarkets. Price discounts on grocery items were not very common on Deliveroo and Uber Eats. Uber Eats does not allow ‘official’ promotions on age-restricted items such as alcohol or tobacco. This is not the case for Deliveroo, which does have price discounts on alcoholic beverages (e.g., 20% off selected wines).

Three convenience stores were found to promote tobacco and e-cigarettes at the top of their online menu. In addition, three virtual alcohol shops on Deliveroo used clear promotional language in the product description of tobacco and e-cigarette products. Quotes include [53,54]:


*“This is one of the best brands of tobacco out there on the market, and has become people’s preferred brand”.*



*“The Marlboro Flavor family, representing quality and tobacco expertise, leads the way in bringing adult smokers the most enjoyable tobacco flavor satisfaction”.*



*“Puff Bar Plus disposable vapes (…) Brand new arrival only on Deliveroo with very cheap price”.*



*“The fantastic minds that created the Geek Bar disposable vapes have made a special flavor for us all here in a Puff Bar form with their Sour Apple hit, this one is a real favorite in house that’s for sure”.*


## 4. Discussion

Though exploratory, this analysis represents the first attempt to assess the new and rapidly evolving environment of dark kitchens and RGDS from a public health perspective. We mapped the breadth of online food delivery in the London Borough of Haringey, ranging from fast food chains with delivery-only dark kitchen locations to virtual alcohol shops located in self-storage facilities.

In line with what is already known about hot food takeaways and online food delivery menus [15,21,22,23,24,25], most identified food businesses available online in Haringey were found to primarily sell fast food or other foods usually high in fat, salt and sugar. As such, the online food environment mirrors, and adds to, the high availability of unhealthy foods already found on many physical high streets, notably high streets in disadvantaged areas [4]. Dark kitchens further increase the geographic availability of predominantly unhealthy food options as most of the (virtual) businesses operating from dark kitchen spaces are not otherwise available in the local area. Virtual food businesses were found to be particularly common as dark kitchens tenants, reflecting the increasing consumer demand for home delivery. Access to a greater amount of (online) food outlets has been associated with increased likelihood of using meal delivery services and could lead to increased consumption of fast food meals [18] and poor health outcomes in the long term [20,25,26,27,28,29]. With their current food offer, dark kitchens may be hypothesized to have an indirect negative influence on public health through this pathway.

Our findings were similar when it comes to RGDS. ‘Unhealthy’ items, including confectionery food items, alcoholic beverages and tobacco products, were prominent on online menus, particularly for alcohol shops, off-licenses and convenience stores available on online delivery platforms. Like dark kitchens, RGDS services make buying grocery items more convenient, while also increasing the temporal availability of these items. This is problematic when many items for sale can contribute to poor health outcomes. It has previously been hypothesized that the increased access to highly processed foods through online grocery delivery services may increase their consumption [9]. Similar pathways could be hypothesized for the consumption of alcoholic beverages. The dark grocer Zapp, which was found to sell the largest number of alcoholic beverages and tobacco products, is ‘open’ 24/7. Virtual alcohol shops are also available for deliveries all night. It is unclear whether online delivery increases access to alcohol and tobacco for minors, especially considering the introduction of ‘contactless deliveries’ during the COVID-19 pandemic. Previous studies found that age verification processes of online alcohol vendors, particularly at point of delivery, were inconsistent and inadequate in preventing sales to minors [55,56].

From the online platforms we viewed we can hypothesize some potential benefits for public health, but also some mechanisms for inequalities. Dark grocer apps and supermarkets available on delivery platforms (but not alcohol shops, off-licenses and convenience stores) increase access to health promoting items like fresh fruit and vegetables and pharmacy items. Grocery delivery services could plausibly benefit consumers who do not have the time or ability to visit physical shops for regular grocery shopping or emergency purchases [9,57], including those with caring responsibilities, mobility issues, and those self-isolating due to COVID-19. However, the higher cost of products and delivery fees may make online grocery deliveries inaccessible for those on lower incomes. Older adults, people with (visual) impairments and those without access to a smartphone or computer may also be less able to access online grocery delivery [9].

It is worrying that online delivery platforms are being used to promote products that are harmful to health. Previous research shows that online food delivery services are more likely to advertise fast food and ultra-processed beverages over healthier options as these tend to be the most popular items [13,14,25,40]. Product imagery was one of the principal ways in which products are promoted in online spaces. This is also true for RGDS, which were found to use similar non-monetary promotion strategies [13,14]. Such strategies have been found effective in influencing costumer purchasing behavior [58] and may increase the urge to buy advertised products impulsively [59]. RGDS were particularly likely to promote alcoholic beverages and instances of tobacco promotion were found on online delivery platforms. The advertising of tobacco products, for example through the use of promotional language is against current tobacco control legislation in the United Kingdom [60], raising the question of whether such platforms and the businesses they partner with are breaking the law or have found a way of circumventing it. In addition, health warnings and clear marking of products containing tobacco and alcohol were inconsistent across the platforms, possibly reflecting the limited regulation of online alcohol and tobacco sales.

### 4.1. Strengths and Limitations

It is clear that the online food environment is still relatively under-researched form a public health perspective, particularly when it comes to more recent developments in online food delivery. This analysis is the first to explore the possible public health impacts of online food delivery through dark kitchens and RGDS by analyzing what health harming and health promoting products they make available to the local population and how. As this work in an initial exploration of one local authority, it was possible to include all local dark kitchens and RGDS. However, this also meant that the analysis was largely descriptive in nature. Due to the rapidly changing online environment, the data that was collected should be seen as a snapshot of dark kitchens and RGDS in one local authority, which might not be representative of the rest of London or England. Furthermore, different parts of the local authority may have slightly different food options available for delivery that were not captured in this analysis. It should be noted that we have focused our analysis on businesses operating in our study area. We do not claim, and have not collected data to establish, that the issues we identify are specific to those businesses or to that area. Researching online food delivery services is complex since the topic involves two intersecting types of environments: the online environment where products are promoted and sold, and the physical environment where products are prepared, stored, delivered and consumed. Our analysis has primarily focused on the online environments. We suggest that future research on this emerging public health issue should focus on more, and larger, jurisdictions and develop methods for integrating assessments of its physical and online dimensions.

### 4.2. Policy Implications

Governments and international organizations are increasingly concerned about the wide availability of unhealthy products in both physical and digital environments [10,15]. While planning restrictions may be successful in controlling the growth of takeaway outlets, for example through restricting fast food outlets near schools [61], such policies do not target online platforms and virtual businesses operating through dark kitchens [18,40]. This is concerning given the generally unhealthy food offer on online food delivery platforms. Online food and grocery delivery services may also undermine alcohol licensing policies. Going forward it will be necessary to find ways to use planning and licensing powers to limit online access to harmful products.

The results of this analysis also highlight the need for improved legislation and enforcement to protect individuals, and especially minors, from viewing health harming promotions when ordering food online. This should be part of a wider policy agenda targeting online marketing of food, alcohol and tobacco, including marketing on social media, which has been advocated as one of the most effective measures against the use of harmful substances [62,63]. Current regulation on how substances like alcohol and tobacco should be displayed and sold online should be updated to ensure that consumers are exposed to health information at point of sale.

Fernandez and Raine argue that marketing and promotion strategies on online food and grocery delivery services can also be used to encourage healthier food choices, for example by recommending foods low in saturated fats or suggesting healthier product swaps [9]. While providing nutrition education and ‘nudging’ interventions could have a positive influence on the choices that consumers make when buying food online [9], this would have to be carried out in collaboration with the online food retailing businesses (who may profit from promoting certain food brands). Local governments could create their own platforms highlighting local food shops that deliver online. These may be better equipped to promote healthy food choices but would likely be more limited in reach.

## 5. Conclusions

To date, public health researchers have not engaged with the health implications of new developments in online food, alcohol and tobacco delivery involving so-called delivery-only ‘dark kitchens’ and rapid grocery delivery services. Our novel analysis into this subject area has therefore been deliberatively explorative in order to highlight areas of concern for future research, policy and practice. The objective of this study was to identify the presence of dark kitchens and RGDS in a metropolitan local authority in London, United Kingdom, and their impact on the availability of food, alcohol and tobacco in the local area. Both dark kitchens and RGDS were found to sell an abundance of food high in fat, salt and sugar, alcohol and/or tobacco products. Given the increased temporal and geographic availability of such products through online delivery services and the convenience of purchasing food online, it is plausible that such services influence the consumption of unhealthy foods and substances. Additional concerns have been identified relating to the promotion of unhealthy products on RGDS, particularly the promotion of alcohol and tobacco. Increased regulation should be in place to limit exposure to such advertising when ordering food online. We encourage further research that expands the geographic area of interest and considers the differences in access and product availability in different areas with varying levels of deprivation.

## Figures and Tables

**Figure 1 ijerph-19-05523-f001:**
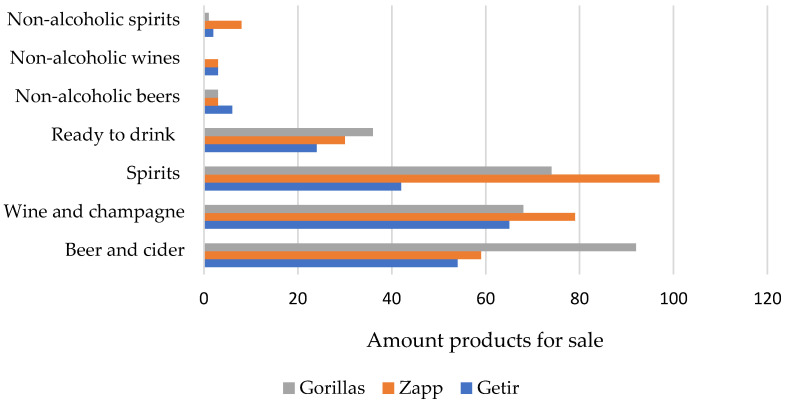
Alcoholic beverages sold by dark grocers [November 2021].

**Figure 2 ijerph-19-05523-f002:**
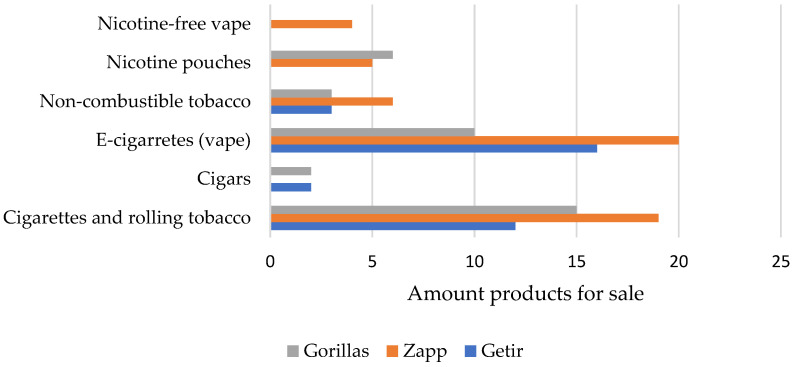
Tobacco and e-cigarette sold by dark grocers [November 2021].

**Table 1 ijerph-19-05523-t001:** Summary information about dark kitchens in the London Borough of Haringey.

Business	Kitchen Units	Food Hygiene Rating [0–5] ^1^	Number of Identified Tenants ^2^	Tenant Types	Equipment, Facilities and Infrastructure	Additional Benefits
Karma Kitchen [48]	Shared workbenches; Private kitchens; Anchor production units [150–900 sq. ft.]	5	72	Food entrepreneurs; National chains; Catering businesses; Virtual brand operators	Kitchen equipment; Extraction and ventilation; Fire rated infrastructure; Waste management; Cleaning services; Pest control	Karma Kitchen Business Support Programme; Product Marketplace; Services Marketplace
Deliveroo Editions [49]	Private kitchens [Size not reported on website]	5	28	Handpicked virtual brand operators; Independent restaurants; and National chains that are already using Deliveroo	Kitchen equipment; Waste management; Cleaning services; Utilities	Marketing support with Growth Manager; Access to Deliveroo data technology; Order and dispatch operations
Foodstars [50]	Private kitchens [200+ sq. ft.]	5	16	Food entrepreneurs; National chains; Catering businesses; Virtual brand operators	Kitchen equipment; Extraction and ventilation; Fire rated infrastructure; Non-slip floor; Cleaning services; Pest control	Central food delivery order processing centre; Delivery driver check-in/management

^1^ Based on the English Food Standards Agency’s Food hygiene ratings (0–5). ^2^ Number of tenants that was identified at the time of data collection (October 2021).

**Table 2 ijerph-19-05523-t002:** Dark grocers in the London Borough of Haringey.

Business	App Name ^1^	Number of Locations in Haringey	Food Hygiene Rating [0–5] ^2^	Products
Getir	Getir: groceries in minutes	4	5	1500+ food items, household supplies, cosmetics, pharmacy items, alcoholic beverages and tobacco
Gorillas	Gorillas: groceries in 10 min	1	5	“Thousands” of food items, household supplies, cosmetics, pharmacy items, alcoholic beverages and tobacco
Zapp	Zapp—24/7 Drinks and Groceries	1	5	“Thousands” of food items, household supplies, cosmetics, pharmacy items, alcoholic beverages and tobacco

^1^ Name of the app on the Apple App Store. ^2^ Based on the English Food Standards Agency’s Food hygiene ratings (0–5).

**Table 3 ijerph-19-05523-t003:** Primary food type or cuisine offered by food outlets operating from dark kitchens in Haringey.

Cuisine or Food Type	Karma Kitchen (*n* = 72)	Deliveroo Editions (*n* = 28)	Foodstars (*n* = 16)	Total (*n* = 116)
British	3	0	0	3 (2.6%)
Burgers	18	5	1	24 (20.7%)
Chicken	8	3	3	14 (12.1%)
Chinese	1	4	0	5 (4.3%)
Dessert	9	3	1	13 (11.2%)
Greek	2	1	0	3 (2.6%)
Indian	3	4	0	7 (6.0%)
Italian	8	2	4	14 (12.1%)
Japanese	1	1	4	6 (5.2%)
Thai	1	1	0	2 (1.7%)
Vietnamese	2	1	1	4 (3.4%)
West African	2	1	0	3 (2.6%)
Other	14	2	2	18 (15.1%)

**Table 4 ijerph-19-05523-t004:** Products shown on the cover images of grocery businesses on delivery platforms. Businesses typically featured more than one product in their cover image.

Promoted Products	Deliveroo	Uber Eats	Total
Alcoholic beverages	33	20	53 (69.7%)
Confectionery	19	5	24 (31.6%)
Cupboard items	12	3	15 (19.7%)
Dairy and eggs	8	1	9 (11.8%)
Fruit and vegetables	5	1	6 (7.9%)
Household supplies	3	1	4 (5.3%)
Meat and alternatives	8	3	11 (14.5%)
Ready meals	8	2	10 (13.2%)
Salty snacks	10	5	15 (19.7%)
Sugar sweetened beverages	12	3	15 (19.7%)
Sweet baked goods	3	0	3 (3.9%)

## Data Availability

Data was obtained from publicly available sources.

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
