# Peer review of "Understanding the Online Environment for the Delivery of Food, Alcohol and Tobacco: An Exploratory Analysis of ‘Dark Kitchens’ and Rapid Grocery Delivery Services"

_ijerph, 2022, doi:10.3390/ijerph19095523_

Round 1

Reviewer 1 Report

I have been very happy reading this text, it is a very original work, and methodologically correct. The only appreciation that I can make is to summarize the text of the results a little more for those data that are collected in the tables.

Author Response

We thank the reviewer for their positive feedback. We have included more information in the text that explains the data summarised in the tables (p. 5, lines 223-226; p. 6, lines 246-248). We now also refer to the tables more clearly throughout the manuscript.

Reviewer 2 Report

The study addresses a topic of interest, the dynamics of this food segment is not often addressed, and its impact has been particularly relevant during the Covid-19 pandemic.  The design and implementation of public policy in the context of the issue is definitely important and the authors should direct their efforts towards a public policy proposal.

The abstract does not make clear the objective of the research and the conclusive aspects of the study are absent, I recommend their incorporation.

Overall, the study is logical and coherent, well structured, even if the research objective needs to be clarified. The conclusion is poor and needs to be further elaborated according to the abundance of data collected and to the objective of the study and the context of its application. The bibliography is poor and obsolete and needs to be expanded and updated.

Author Response

We thank the reviewer for their suggestions. We have drafted a report with policy recommendations for the local authority that was analysed in our study.

We have reviewed the abstract to more clearly include the research objective and concluding remarks (p.1). We have also adapted the wording in section ‘1.3 Scope of this paper’ (p.3) to clarify our research objective: ”The aim is to understand and assess the availability of health harming and health promoting products through dark kitchens and RGDS. We assessed where these services are located, how these services work, and how they impact on the local availability of products known to affect health, focusing primarily on food, alcoholic and non-alcoholic beverages and tobacco”

As suggested, the conclusion has been rewritten to include the study objective and a short summary of findings (p. 15, lines 614-632).

We have updated the bibliography with the following studies on among others the digital food environment and the links between food outlet exposure and health outcomes:

  1. Bandyopadhyay, N.; Sivakumaran, B.; Patro, S.; Kumar, R.S. Immediate or delayed! Whether various types of consumer sales promotions drive impulse buying?: An empirical investigation. Journal of Retailing and Consumer Services. 2021, 61, 102532, doi: 10.1016/j.jretconser.2021.102532.
  2. Bennett, R.; Zorbas, C.; Huse, O.; Peeters, A.; Cameron, A.J.; Sacks, G.; Backholer, K. Prevalence of healthy and unhealthy food and beverage price promotions and their potential influence on shopper purchasing behaviour: A systematic review of the literature. Obesity Reviews. 2020, 21, doi: 10.1111/obr.12948.
  3. Brown, H.; Kirkman, S.; Albani, V.; Goffe, L.; Akhter, N.; Hollingsworth, B.; von Hinke, S.; Lake, A. The impact of school exclusion zone planning guidance on the number and type of food outlets in an English local authority: A longitudinal analysis. Health & Place. 2021, 70, 102600-102600, doi:10.1016/j.healthplace.2021.102600.
  4. Burgoine, T.; Forouhi, N.G.; Griffin, S.J.; Wareham, N.J.; Monsivais, P. Associations between exposure to takeaway food outlets, takeaway food consumption, and body weight in Cambridgeshire, UK: population based, cross sectional study. BMJ : British Medical Journal. 2014, 348, g1464, doi:10.1136/bmj.g1464.
  5. Donin, A.S.; Nightingale, C.M.; Owen, C.G.; Rudnicka, A.R.; Cook, D.G.; Whincup, P.H. Takeaway meal consumption and risk markers for coronary heart disease, type 2 diabetes and obesity in children aged 9-10 years: a cross-sectional study. Archives of Disease in Childhood. 2018, 103, 431-436.
  6. Fernandez, M.A.; Raine, K.D. Digital Food Retail: Public Health Opportunities. Nutrients. 2021, 13, 3789, doi:10.3390/nu13113789.
  7. Giskes, K.; van Lenthe, F.; Avendano-Pabon, M.; Brug, J. A systematic review of environmental factors and obesogenic dietary intakes among adults: are we getting closer to understanding obesogenic environments? Obesity Reviews. 2011, 12, e95-e106, doi:https://doi.org/10.1111/j.1467-789X.2010.00769.x.
  8. Granheim, S.I.; Løvhaug, A.L.; Terragni, L.; Torheim, L.E.; Thurston, M. Mapping the digital food environment: A systematic scoping review. Obesity Reviews. 2022, 23. doi: 10.1111/obr.13356.
  9. Halloran, A.; Faiz, M.; Chatterjee, S.; Clough, I.; Rippin, H.; Farrand, C.; Weerasinghe, N.; Flore, R.; Springhorn, H.; Breda, J.; et al. The cost of convenience: potential linkages between noncommunicable diseases and meal delivery apps. The Lancet Regional Health Europe. 2022, 12, doi:10.1016/j.lanepe.2021.100293.
  10. Jia, S.S.; Gibson, A.A.; Ding, D.; Allman-Farinelli, M.; Phongsavan, P.; Redfern, J.; Partridge, S.R. Perspective: Are Online Food Delivery Services Emerging as Another Obstacle to Achieving the 2030 United Nations Sustainable Development Goals? Frontiers in nutrition. 2022, 9, 858475-858475. doi:10.3389/fnut.2022.858475.
  11. Kant, A.K.; Whitley, M.I.; Graubard, B.I. Away from home meals: associations with biomarkers of chronic disease and dietary intake in American adults, NHANES 2005–2010. International Journal of Obesity. 2015, 39, 820-827, doi:10.1038/ijo.2014.183.
  12. Li, C.; Mirosa, M.; Bremer, P. Review of Online Food Delivery Platforms and their Impacts on Sustainability. Sustainability. 2020, 12, doi:10.3390/su12145528.
  13. Maguire, E.R.; Burgoine, T.; Penney, T.L.; Forouhi, N.G.; Monsivais, P. Does exposure to the food environment differ by socioeconomic position? Comparing area-based and person-centred metrics in the Fenland Study, UK. International Journal of Health Geographics. 2017, 16, 33, doi:10.1186/s12942-017-0106-8.
  14. Smith, K.J.; Blizzard L Fau - McNaughton, S.A.; McNaughton Sa Fau - Gall, S.L.; Gall Sl Fau - Dwyer, T.; Dwyer T Fau - Venn, A.J.; Venn, A.J. Takeaway food consumption and cardio-metabolic risk factors in young adults. European Journal of Clinical Nutrition. 2012, 66.
  15. Wellard-Cole, L.; Davies, A.; Allman-Farinelli, M. Contribution of foods prepared away from home to intakes of energy and nutrients of public health concern in adults: a systematic review. Critical Reviews in Food Science and Nutrition. 2021, 1-12, doi:10.1080/10408398.2021.1887075.
  16. World Health Organization. Slide to order: a food systems approach to meals delivery apps: WHO European Office for the Prevention and Control of Noncommunicable diseases. 2021

We welcome any further literature suggestions.

Reviewer 3 Report

Please to refer at the similar studies.
Yours methodologies validated by other authors.

Author Response

We thank the reviewer for their suggestion. We now reference other studies on online food delivery that used similar data collection methods (p.4, 149-150).

Horta, P.A.-O.; Matos, J.P.; Mendes, L.L. Digital food environment during the coronavirus disease 2019 (COVID-19) pandemic in Brazil: an analysis of food advertising in an online food delivery platform. British Journal of Nutrition 2021, 126, 767-772, doi:10.1017/S0007114520004560.

Partridge, S.A.-O.; Gibson, A.A.-O.; Roy, R.A.-O.; Malloy, J.A.; Raeside, R.; Jia, S.S.; Singleton, A.A.-O.; Mandoh, M.A.-O.; Todd, A.R.; Wang, T.; et al. Junk Food on Demand: A Cross-Sectional Analysis of the Nutritional Quality of Popular Online Food Delivery Outlets in Australia and New Zealand. Nutrients 2020, 12, 3107, doi:10.3390/nu12103107.

Horta, P.M.; Souza, J.P.M.; Rocha, L.A.-O.; Mendes, L.A.-O. Digital food environment of a Brazilian metropolis: food availability and marketing strategies used by delivery apps. Public Health Nutrition 2021, 24, 544-548, doi:10.1017/S1368980020003171.

Reviewer 4 Report

Dear Authors,

the paper “Understanding the Online Environment for the Delivery of Food, Alcohol and Tobacco: An Exploratory Analysis of ‘Dark kitchens’ and Rapid Grocery Delivery Services” is an interesting study with the intent to help in the assessment of the rapidly evolving environment of online delivery services. It is well written and of high interest to the Public Health community. This environment has especially grown during the pandemic area of the last two years, while Public Health Authorities were occupied by other challenges. Therefore the need to asses these environments from a consumer safety perspective is highly relevant. Although this study is limited to one local authority it nevertheless helps Public health researchers to start to understand the mechanisms and challenges.The analysis obviously followed a well-planned and structured design.  

Author Response

We thank the reviewer for their positive feedback and hope that this study encourages future research on the topic.

Reviewer 5 Report

The scientific article deals with the hitherto little-studied topic - "dark kitchens" and RGDS. It is mentioned in the introductory part that the aim of the article is to understand how online food delivery works: "dark kitchens" and RGDS, which would lead to proposals for public health policy development. Unfortunately, at the end of the article, the authors do not make any recommendations or suggestions that could be implemented to achieve the above goal.

It is mentioned in section 4.1, lines 514-515, that this article explains the impact of dark kitchens and RGDS on public health, however, both in the analysis of the results and in the discussion section, the link between public health and the research objects is not really reflected.
For example, in section 3.2, a map could be created that would show the results in a much easier way for the reader to understand.

The article would need to be supplemented with scientific justification. The authors emphasize that research on dark kitchens and RGDS is relatively limited, but the article aims to look at this topic from a public health perspective. This aspect should be included in the article.

To supplement the scientific section of the article by showing a broader link between public health and the research topic.

Author Response

We thank the reviewer for their useful suggestions.

We have added an additional section to the discussion highlighting some policy suggestions (Section 4.2, p.14, lines 559-586). These suggestions have been discussed with senior public health, environmental health and other relevant practitioners working at the local authority that was analysed.

As suggested, we have made the links between our study objective and public health more clear in the manuscript, particularly in the introduction and discussion. We have for example specified the possible direct and indirect health impacts of online food delivery on page 2 (lines 67-71): “Similar to ‘fast food’ take-away meals, meals ordered online tend to be high in fat, salt and sugar and low in whole grains, fruits and vegetables [15,21-25]. Frequent consumption of these types of meals has been associated with poorer diets [25-27], greater odds of overweight and obesity [20,25] and higher cardio-metabolic [28] and cardio-vascular risk factors [27,29]. Concerns have also been raised about the impacts of food delivery on the environment (ie. high carbon footprint and waste generation) and traffic systems (ie. traffic congestion, road accidents), which could affect public health indirectly [30,31].

The health impacts of dark kitchens and RGDS have also been made more explicit in the discussion on p. 13, lines 514-517; lines 522-526 and p.14, lines 549-552. The highlighted section 4.1 on page 14 now specifies how our analysis links to public health: “This analysis is the first to explore the possible public health impacts of online food delivery through dark kitchens and RGDS by analysing what health harming and health promoting products they make available to the local population and how.”

We have also included further information on the concerns that have been raised by public health researchers (p. 3, lines 102-104, p.14, lines 589-591).

We are wary of including a map in section 3.2 because we do not report on any specific GIS analysis on delivery radii, and so the map would just re-state a simple point already made in the text (that there are dark kitchens and RGDS in the study area). As part of our stakeholder consultations we produced a map to present to the local practitioners who worked in that area. This had meaning to them because they knew and worked in that area. However, we do not think a general audience who are unfamiliar with the locality would get as much from it. We have gone back to section 3.2 and made changes that we hope will increase clarity for the reader (p. 7-8).